# Tackling realistic Li$^+$ flux for high-energy lithium metal batteries

Shuoqing Zhang[1,7], Ruhong Li[1,7], Nan Hu [2,7], Tao Deng[3], Suting Weng[4], Zunchun Wu[1], Di Lu[1], Haikuo Zhang[1], Junbo Zhang[1], Xuefeng Wang [4], Lixin Chen [1,5], Liwu Fan[2,6] ✉ & Xiulin Fan[1] ✉

Electrolyte engineering advances Li metal batteries (LMBs) with high Coulombic efficiency (CE) by constructing LiF-rich solid electrolyte interphase (SEI). However, the low conductivity of LiF disturbs Li$^+$ diffusion across SEI, thus inducing Li$^+$ transfer-driven dendritic deposition. In this work, we establish a mechanistic model to decipher how the SEI affects Li plating in high-fluorine electrolytes. The presented theory depicts a linear correlation between the capacity loss and current density to identify the slope $k$ (determined by Li$^+$ mobility of SEI components) as an indicator for describing the homogeneity of Li$^+$ flux across SEI, while the intercept dictates the maximum CE that electrolytes can achieve. This model inspires the design of an efficient electrolyte that generates dual-halide SEI to homogenize Li$^+$ distribution and Li deposition. The model-driven protocol offers a promising energetic analysis to evaluate the compatibility of electrolytes to Li anode, thus guiding the design of promising electrolytes for LMBs.

The revived Li metal batteries (LMBs) pave the way to the target energy density of >350 Wh kg$^{-1}$ thanks to Li metal anode (LMA) with the highest theoretical specific capacity (3860 mAh g$^{-1}$) and the lowest redox potential (−3.04 V vs. the standard hydrogen electrode) among all possible anodes[1–3]. However, dendritic Li and low Coulombic efficiency (CE) deteriorate LMBs. This is mainly attributed to the absence of a stable and uniform solid electrolyte interface (SEI) dictated by the interfacial reactions between the LMA and electrolytes[4–6]. An ideal SEI should hold the merits of fast Li$^+$ but negligible electron conduction, high mechanical strength, and high interfacial energy to LMA[7]. Therefore, electrolyte engineering is decisive in inhibiting Li dendrites and realizing high CE by tuning the SEI components.

LiF has been regarded as one of most effective SEI components due to its low electronic conductivity and high surface energy (73.28 meV Å$^{-2}$)[8], which can prevent the formation of Li/SEI interface

(i.e., Li dendrites). Moreover, the small lattice constant of LiF allows the SEI to deform elastically with a constantly changing morphology of LMA[9]. Hence, constructing LiF-rich SEI shows effectiveness in suppressing Li dendrites and preventing side reactions between LMA and electrolytes[10–15]. Inspired by this concept, a myriad of efforts have been devoted to modulating fluorinated electrolytes, including fluorinated solvents[16–21], electrolyte additives[22–25], high-concentration electrolytes (HCE)[9,26–28] and localized HCE[29–33], etc. These electrolytes succeeded in building LiF-rich SEI due to their high-fluorine content, which enables reversible LMBs featuring impressive CE values of >99%. However, LiF suffers from poor Li$^+$ conductivity (~ 10$^{-31}$ S cm$^{-1}$)[34], i.e., a high Li$^+$ diffusion energy barrier, which can cause inhomogeneous Li$^+$ flux across SEI. The uneven Li$^+$ distribution at the substrate surface could induce undesired dendritic deposition as the cycle proceeds[35]. This kinetic mechanism of Li dendrite formation in LMBs remains unsolved despite the aforementioned advantages of LiF-rich SEI. Thus, revealing how the

[1]State Key Laboratory of Silicon Materials, School of Materials Science and Engineering, Zhejiang University, Hangzhou 310027, China. [2]State Key Laboratory of Clean Energy Utilization, School of Energy Engineering, Zhejiang University, Hangzhou 310027, China. [3]Department of Chemical and Biomolecular Engineering, University of Maryland, College Park, MD 20742, USA. [4]Beijing National Laboratory for Condensed Matter Physics, Institute of Physics, Chinese Academy of Sciences, Beijing 100190, China. [5]Key Laboratory of Advanced Materials and Applications for Batteries of Zhejiang Province, Hangzhou 310013, China. [6]Key Laboratory of Clean Energy and Carbon Neutrality of Zhejiang Province, Hangzhou 310027, China. [7]These authors contributed equally: Shuoqing Zhang, Ruhong Li, Nan Hu. ✉e-mail: liwufan@zju.edu.cn; xlfan@zju.edu.cn

SEI kinetically affects Li deposition is highly demanded for designing advanced electrolytes.

As an early model referring to transition metal deposition in aqueous solutions, Sand's time ($t_{Sand}$) recurs to describe the onset of dendritic Li growth[36,37]. The $t_{Sand}$ features a zero Li$^+$ concentration at the substrate surface. The cation-deficient zones promote Li growth at surface protrusions, which quickly develop into sharp dendrites due to the continuously preferential deposition. Multiple studies have proposed some underlying Li growth modes inspired by $t_{Sand}$, which suggested significant strategies for more durable LMBs[38–41]. It should be noted that $t_{Sand}$ focuses on the Li$^+$ transfer through bulk electrolyte while omits the subsequent Li$^+$ migration inside SEI, which has been considered the rate-limiting step for Li deposition[42]. Additionally, the use of $t_{Sand}$ requires that the actual current density reaches or exceeds the limited value. This is inaccessible in practical LMBs because the short inter-electrode distance defines a high threshold of 250 mA cm$^{-2}$ [43]. Therefore, the modeling of Li growth in actual cases is still poorly developed.

In this work, to address the above challenges, we establish a mechanistic protocol that deciphers the dependence of Li deposition on SEI, validated by an explicit assessment reflecting the compatibility of the most successful fluorine-rich electrolytes to LMA. The jagged Li deposition originates from the non-uniform Li$^+$ mobility of SEI components. A promising strategy to accommodate uniform Li$^+$ distribution over the substrate is enhancing Li$^+$ conductivity of LiF regions in SEI. Such implications of the proposed protocol inspire the design of a dual-halide (F and Cl) electrolyte, which in situ produces a dual-halide (LiF$_{1-x}$Cl$_x$) SEI on LMA.

Compared to the LiF phase, Cl doping enables the LiF$_{1-x}$Cl$_x$ phase to have a fast Li$^+$ conductivity together with a six-fold lower energy barrier without compromising mechanical stability. The effectiveness is evidenced by an improved CE (>99.5%) in Li||Cu cells and prolonged cycle life (>200 cycles) in full cells. Specially, anode-free Cu||LiNi$_{0.5}$Co$_{0.2}$Mn$_{0.3}$O$_2$ pouch cells with the dual-halide electrolyte realize >125 cycles at practical levels. The proposed protocol enables fundamental understanding and evaluation of Li deposition and opens up a feasible engineering approach for realizing high-energy LMBs.

## Results

### Establishment and application of Li deposition model

As shown in Fig. 1a, the major SEI components can be classified into two groups with high and low Li$^+$ mobility, according to their distinct energy barriers for Li$^+$ diffusion. Hence, the Li deposition process is influenced by the local energy barrier of SEI, accompanied by the inhomogeneous distribution of electrolyte concentration. Although SEI has a complex composition and distribution of components, it can be simply distinguished into high and low mobility zones by equivalence approximation (Fig. 1b). To quantitatively assess how the SEI affects Li deposition, we establish a model based on the law of Li mass conservation (Fig. 1c). Figure 1d displays an equivalent circuit (Detailed discussion in Supplementary Note 1) to elucidate the effects of various parameters on Li$^+$ diffusion across electrolyte and SEI. The total Li$^+$ capacity $Q_{total}$ of an LMA can be divided into irreversible loss $Q_{ir}$ due to dead Li and SEI formation, residual Li $Q_{Li-residue}$

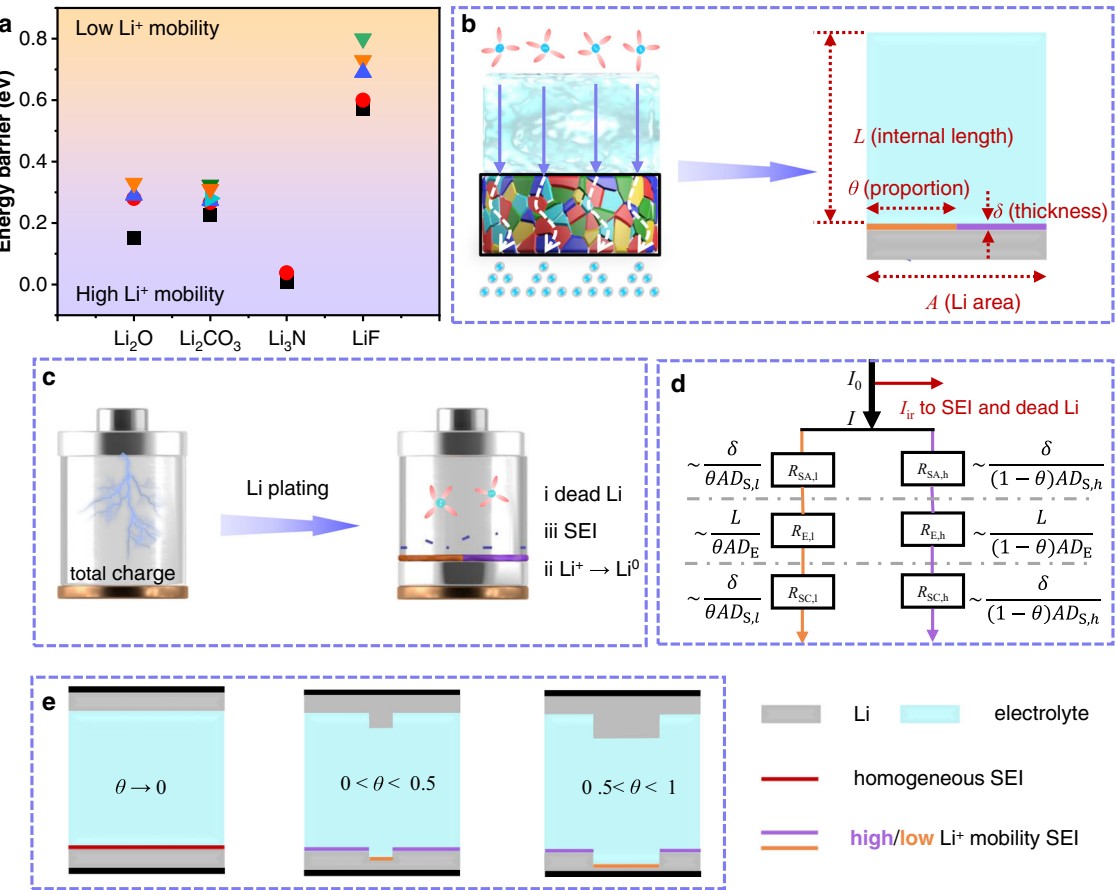

**Fig. 1 | Model of Li deposition in LMBs. a** Diffusion energy of Li$^+$ diffusion for various SEI components[34,66–69]. **b** Schematic illustration of Li deposition process and model parameters. **c** Capacity conservation during a complete Li deposition process. **d** Equivalent circuit for Li$^+$ diffusion across the electrolyte and SEI ($R_{SA}$ and $R_{SC}$ represent the anode and cathode interface resistance, respectively). **e** Dependence of Li plating on the Li$^+$ mobility of SEI.

(Supplementary Note 2) due to uneven deposition and desired Li deposition $Q_{deposit}$, i.e.

$$Q_{total} = Q_{deposit} + Q_{ir} + Q_{Li-residue} \qquad (1)$$

An evaluation parameter $Q_{loss}$ can be readily defined as

$$Q_{loss} = Q_{ir} + Q_{Li-residue} = Q_{ir} + \frac{t_{dis}(j_h - j_l)A\theta}{nF} \qquad (2)$$

where $j_h$ and $j_l$ represent the current density corresponding to high and low mobility pathways, respectively, $t_{dis}$ is total deposition time, $A$ is the area of Li foil, $\theta$ means the proportion of low mobility region, $n$ is the stoichiometric number of electrons consumed in the electrode reaction (e.g., 1 for reduction of Li$^+$) and $F$ is the Faraday's constant (96485 C mol$^{-1}$).

Then, Eq. (2) can be rearranged after substituting Eq. (10) in supporting information

$$Q_{loss} = Q_{ir} + \frac{At_{dis}j}{nF} \cdot \frac{1 - \frac{D_{s,l}}{D_{s,h}}}{\frac{(1-\theta)}{\theta} + \frac{D_{s,l}}{D_{s,h}} + \frac{LD_{s,l}}{2\theta\delta D_E}} = Q_{ir} + k \cdot j \qquad (3)$$

where a slope $k$ is introduced for simplifying the linear expression, $L$ is the internal electrode distance, $\delta$ is the thickness of SEI, $D_E$ represents the Li$^+$ diffusion in a bulk electrolyte, $D_{s,l}$ and $D_{s,h}$ represents low and high Li$^+$ diffusion through SEI, respectively.

It is worth noting that $t_{dis}$ is determined by the total capacity and applied current density together and will be a specific constant value under a certain condition. As for the slope $k$, it is a significant parameter over the range from 0 to 1 that reflects the homogeneity of Li$^+$ flux across SEI. The detailed $k$ value can be influenced by several factors but mainly by the Li$^+$ mobility of SEI components: (i) Initial roughness of Li foil and separators can disturb Li$^+$ diffusion pathways; (ii) Viscosity and conductivity of electrolytes can affect Li$^+$ diffusion velocity; (iii) Difference between $D_{s,l}$ and $D_{s,h}$ takes the major responsibility for uneven Li$^+$ distribution before Li deposition. A homogenous diffusion across SEI will be realized when '$D_{s,l} \rightarrow D_{s,h}$; or '$\theta \rightarrow 0$' (Fig. 1e), which also means '$k \rightarrow 0$'. The larger the $k$ deviates from 0, the more heterogeneous the Li$^+$ flux is. Moreover, a larger proportion of low mobility SEI, i.e., higher $\theta$, leads to larger $k$ as well as more Li-residual capacity loss. The low utilization of Li foil will undermine LMBs because a thin Li foil or zero excess Li is always required to maximize the energy density. Additionally, the intercept $Q_{ir}$ indicates the irreversible capacity due to the formation of SEI or dead Li. Thus, the maximum $CE$ of LMA in a selected electrolyte can be determined by:

$$CE_{max} = \frac{Q_{total} - Q_{ir}}{Q_{total}} \qquad (4)$$

Therefore, Eqs. (3 and 4) offer a methodology to evaluate the electrochemical performance of LMA in a designed electrolyte. Different from Sand's time which focuses only on the bulk electrolyte, our proposed model integrates the SEI properties with bulk electrolyte to manifest critical parameters for Li growth.

To validate the proposed theory, the most efficient electrolytes (Table S1) reported recently were employed for the investigation based on Eq. (3) (Details in Fig. S3). The relationships of $Q_{loss}$ vs. $j$ for different electrolytes are displayed in Fig. 2a. All the fitted plots present an obvious linear correlation, demonstrating the feasibility of this mathematical model in evaluating different electrolytes. Moreover, the potential $CE_{max}$ of LMA in various electrolytes is evaluated by Eq. (4). The obtained $k$, $Q_{ir}$, and $CE_{max}$ are presented in Fig. 2b. HCE, dimethyl carbonate-1,1,2,2-tetrafluoroethyl-2,2,3,3-tetrafluoropropyl      ether

(DMC-TTE), dimethoxyethane-fluorobenzene (DME-FB) and DME-TTE show high $k$ values (15.410, 9.289, 7.072, 3.007). Although these advanced electrolytes have shown high CE[26,30,31,44], the LiF-rich SEI with a high energy barrier still leads to inhomogeneous Li$^+$ distribution at high Li plating capacity. It should be noted that the decreasing order of $k$ values follows the increasing order of Li$^+$ conductivity of the electrolytes (Fig. S4a). This agrees well with the empirical rule that electrolytes with higher bulk ionic conductivity often generate SEI with lower impedance[5,45]. Moreover, the high viscosity of HCE (Fig. S4b) further increases the $k$ value (15.410). BE exhibits a low $k$ (1.742) due to its high Li$^+$ conductivity, but the high $Q_{ir}$ (3.825 mAh cm$^{-2}$) suggests a low CE for LMA. The delicate SEI and dead Li formed in BE exclude its application in LMBs[46]. Therefore, enhancing the Li$^+$ conductivity of LiF-rich SEI without compromising the mechanical strength is promising to stabilize LMA. To this end, a dual-halide electrolyte (1.3 M LiFSI in DME/1,2-dichloroethane (DCE) shown in Table S2 and Fig. S4c, termed as 1.3 M LDC) is specially designed to produce dual-halide (LiF$_{1-x}$Cl$_x$) SEI (Fig. S5), where Cl doping can endow the LiF$_{1-x}$Cl$_x$ phase with fast Li$^+$ conductivity and sufficient mechanical stability due to the lower ionic migration energy barrier (LiCl vs. LiF, 0.09 eV vs. 0.17 eV)[47,48] and high surface energy (37.55 meV Å$^{-2}$)[8] of LiCl (This will be discussed in detail later). As shown in Fig. 2a, b, 1.3 M LDC shows the lowest $k$ value (0.533) among all the electrolytes, manifesting that the dual-halide SEI can support uniform Li$^+$ diffusion and maintain stable Li growth at various current densities. Furthermore, the lowest $Q_{ir}$ (0.232 mAh cm$^{-2}$) and highest $CE_{max}$ (99.75%) indicate the impressive electrode/electrolyte interface chemistry in LDC electrolyte.

To highlight the reliability of dual-halide electrolytes on stable Li plating, the optical images of Li deposits are displayed in Fig. 2c, which reconfirm the schematic models in Fig. 1e. Both BE and HCE electrolytes lead to rough Li deposits and obvious Li residues. Flat Li deposits and clean shells are observed in 1.3 M LDC electrolyte, which remains consistent even at high current densities (Fig. S6). Therefore, the fluctuation of local current density rather than high average current density induces Li dendrites[43]. According to the proposed protocol, LiF-rich SEI with $k > 0$ suffers from inhomogeneous Li$^+$ diffusion, promoting the appearance of Li dendrites (Fig. 2d). The LiF$_{1-x}$Cl$_x$-rich SEI lowers the $k \rightarrow 0$ because of the low and homogeneous energy barrier for Li$^+$ diffusion. To clarify this principle, the Li$^+$ flux and potential drop across electrolyte and SEI are visualized, respectively. The high Li$^+$ diffusion energy barrier of LiF-rich SEI is primarily responsible for the uneven Li$^+$ concentration across electrolytes (Fig. S7) and SEI (Fig. 2e, f) before Li deposition. However, the high Li$^+$ mobility of LiF$_{1-x}$Cl$_x$-rich SEI enables uniform Li flux and potential distribution through both electrolytes (Fig. S8) and SEI (Fig. 2g, h), realizing high-efficiency Li plating/stripping.

## Interface chemistry of LMA in dual-halide electrolyte

To elucidate the interfacial chemistry of LMA in the dual-halide electrolyte, Li$^+$ solvation structure and surface components are investigated to clarify the formation of dual-halide SEI on LMA. Figure 3a displays the Raman spectra of different electrolytes. Free DME molecules are characterized by peaks at 820 and 847 cm$^{-1}$[44]. As the Li$^+$ concentration increases, the free DME molecules are coordinated by the Li$^+$ ions, with the peak shifting to 872 cm$^{-1}$ in HCE. Meanwhile, the free FSI$^-$ anions at 717 cm$^{-1}$ blueshifts to 752 cm$^{-1}$, which indicates that the FSI$^-$ anions are also involved in the Li$^+$ solvation structure in the form of contact ion pairs (CIPs) or aggregate (AGG)[49,50]. With the addition of DCE, the solvation structures remain unchanged. To further specify the Li$^+$ solvation structure, molecular dynamics (MD) simulation of 1.3 M LDC electrolyte was conducted (Fig. 3b). The solvation shell of Li$^+$ ions was statistically analyzed, as displayed in Fig. 3c. In the Li$^+$ solvation shell, the ratio of FSI$^-$, DME and DCE is 2.67:1.02:0.02 on average. In detail, FSI$^-$/DME with the statistical ratio of 3/1 and 2/1 accounts for 45 and 29%, respectively. The representative

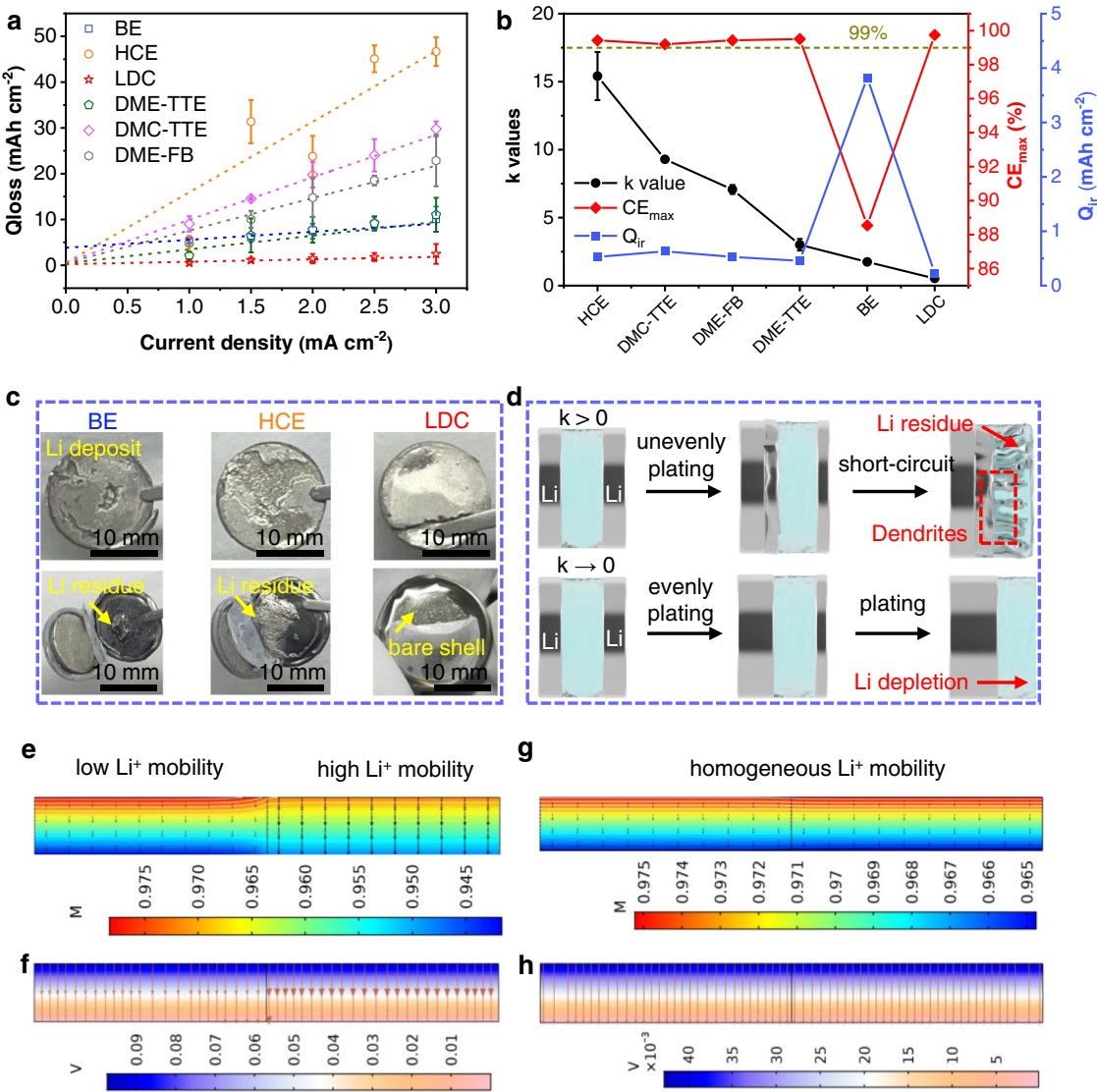

**Fig. 2 | Modeling of Li plating behaviors in different electrolytes. a** Fitted plots of $Q_{loss}$ vs. $j$ based on Eq. (3). **b** Comparison of $k$ values, $CE_{max}$, and $Q_{ir}$ for different electrolytes. **c** Optical images of Li deposits and cathode shells in BE, HCE, and 1.3 M LDC, respectively. **d** Schematic illustration of Li deposition in the case of $k > 0$ and $k \to 0$. The simulation results of $Li^+$ concentration and potential distribution across the LiF-rich SEI (**e, f**) and dual LiF$_{1-x}$Cl$_x$-rich SEI (**g, h**). All error bars are evaluated by standard deviation.

solvation structures are illustrated in Fig. 3d. When the statistics are centered on the FSI⁻ anions, the number of adjacent Li⁺ ions above 2 accounts for 90% (Fig. 3e). Furthermore, the radial Li-Li pair distribution function was calculated and analyzed in Fig. S9, where ion clusters with a size of 6 Å account for the largest proportion. These results demonstrate the AGG solvation structure dominates in 1.3 M LDC electrolyte. The radial distribution functions and corresponding coordination number of Li-O$_{DME}$, Li-O$_{FSI}$, and Li-Cl$_{DCE}$ pairs were calculated from the final 1 ns trajectory, as shown in Fig. 3f. The sharp peaks at 2 Å suggest the close contact of Li⁺/DME and Li⁺/FSI⁻ pairs, while the weak hump at 6.5 Å of Li-Cl$_{DCE}$ pair indicates the feeble interactions between Li⁺ ions and DCE molecules. The weak solvation of DCE molecules to Li⁺ ions is also observed in the snapshots of simulated 1.3 M LDC electrolyte (Fig. 3b). These phenomena indicate the preferential decomposition of FSI⁻ anions in 1.3 M LDC electrolyte, accompanied by the DCE decomposition (Fig. S5) to produce LiF$_{1-x}$Cl$_x$ species, which is demonstrated by the ab initio MD in Fig. S10.

The interfacial components on LMA were further determined by X-ray photoelectron spectroscopy (XPS) with different sputtering time. For LMA cycled in 1.3 M LDC electrolyte, the LiF is originated from the FSI⁻ decomposition (Fig. 3g), associated with the formation of C-SO$_x$ species in C 1$s$ spectra (Fig. 3i). There are also partial Cl-related species as guest halide components in SEI (Fig. 3h). The dual-halide SEI can be further evidenced by the LiX (X = F and Cl) species in Li 1$s$ spectra (Fig. 3j)[48]. Therefore, the SEI formed in 1.3 M LDC is based on the FSI⁻ decomposition and aided by DCE molecules, yielding dual-halide LiF$_{1-x}$Cl$_x$-rich SEI. By contrast, the SEI formed in BE presents obvious PO$_x$F$_y$ compounds resulted from PF$_5$ or PF$_6$⁻ decomposition (Fig. S11), which inevitably produces corrosive HF[5]. Moreover, the existence of poly(CO$_3$) species indicates significant decomposition of solvents, leading to a less protective SEI on LMA. Although inorganic LiF emerges from anion decomposition, a large proportion of organic species tends to impair the mechanical stability of SEI[51].

A theoretical simulation was conducted to reveal how the dual-halide SEI modifies Li⁺ diffusion in SEI. Herein, diversified SEI components are investigated, including Li$_2$O (Fig. S12), Li$_2$CO$_3$ (Fig. S13), Li$_3$N (Fig. S14), LiF (Fig. S15), and LiF$_{1-x}$Cl$_x$. According to the binding energy landscape for Li⁺ migration (Figs. S12c, S13c, S14c, 4a), the energy barrier follows the order of Li$_3$N < Li$_2$O < Li$_2$CO$_3$ < LiF, indicating the LiF region limits the fast Li⁺ transferring. For building the LiF$_{1-x}$Cl$_x$ model,

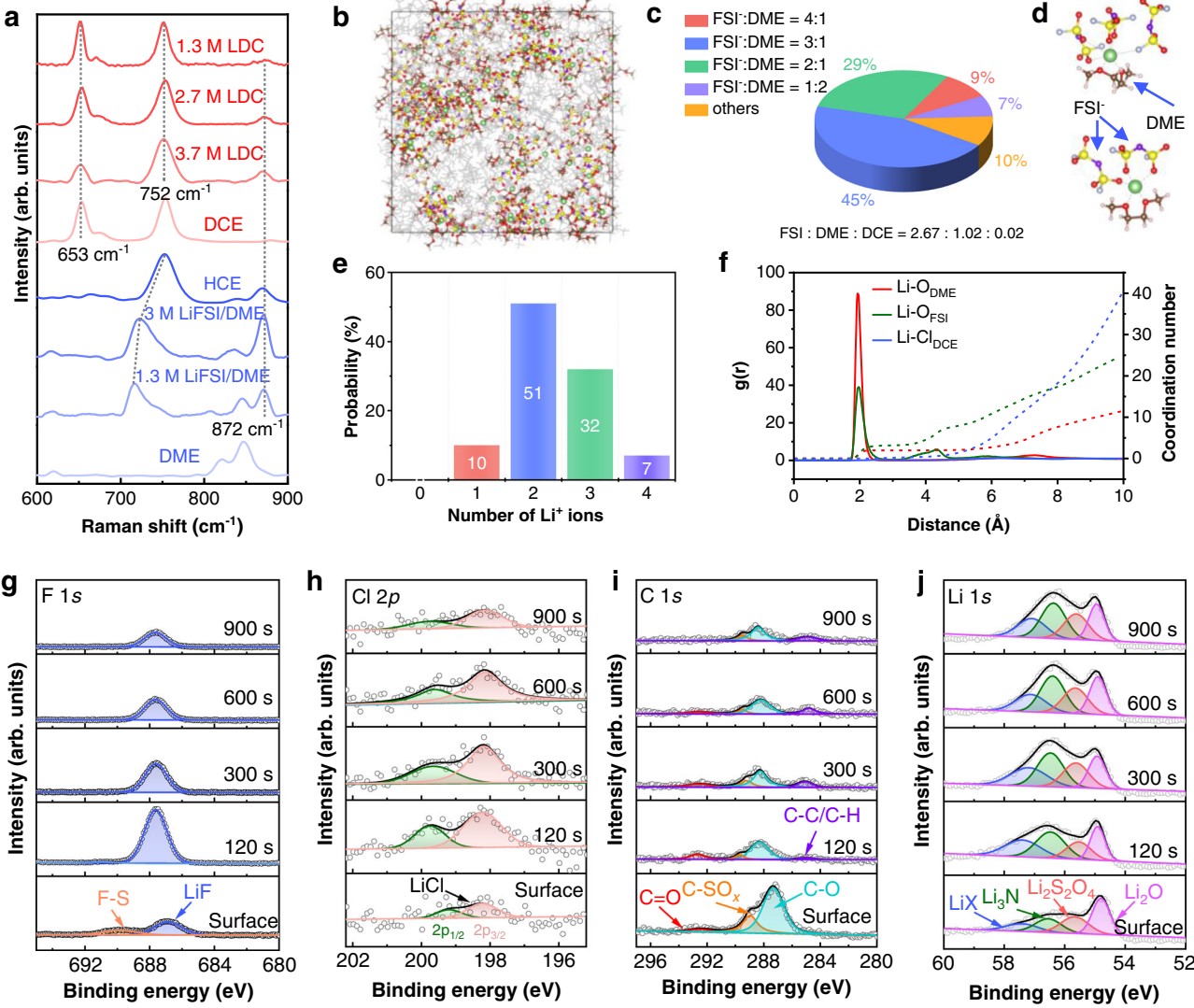

**Fig. 3 | Analysis of Li⁺ solvation structures and SEI components. a** Raman spectra of the solvents and electrolytes. **b** Simulated structures of 1.3 M LDC electrolyte. **c** Proportion of FSI⁻/DME with different ratios in 1.3 M LDC electrolyte. **d** Typical Li⁺ solvation structures with FSI⁻/DME ratio of 3/1 and 2/1. **e** Number of adjacent Li⁺ ions centered on the FSI⁻ anions. **f** Radial distribution functions of Li-O$_{DME}$, Li-O$_{FSI⁻}$, and Li-Cl$_{DCE}$ pairs in 1.3 M LDC electrolyte. XPS depth profiles of **g** F 1$s$ spectra, **h** Cl 2$p$ spectra, **i** C 1$s$ spectra, and **j** Li 1$s$ spectra. The LMA is obtained from a Li||Cu cell, which is cycled at 0.5 mA cm⁻² with a fixed capacity of 4 mAh cm⁻².

Cl content was determined to be ~10% according to the XPS results (Fig. S16). Possible configurations of LiF$_{1-x}$Cl$_x$ were constructed and optimized to screen out the unstable states, twenty of which were shown in Fig. S17. According to the formation energy and energy above hull summarized in Fig. S17u, the LiF$_{1-x}$Cl$_x$ configuration in Fig. S17t is the most stable state, in which some of F atoms are randomly replaced by Cl atoms. Fig. 4a, b illustrate the binding energy landscape when Li⁺ ions hop at the surface of LiF and LiF$_{1-x}$Cl$_x$. Compared to the bare LiF, the introduced Cl atoms enlarge the regions with low binding energy. Moreover, the contour lines of LiF$_{1-x}$Cl$_x$ are more continuous and flatter, which is favorable for fast Li⁺ transport. Two Li⁺ diffusion pathways are identified according to the binding energy landscape (Fig. 4c). These two energy barriers of 0.18 and 0.23 eV for Li⁺ diffusion along the LiF grain boundaries are significantly reduced to 0.03 and 0.09 eV after the Cl⁻ doping (Fig. 4d), respectively. The energy barrier of the preferred path 1 is reduced by a factor of six. To further unveil the Li⁺ diffusion through bulk LiF and LiF$_{1-x}$Cl$_x$, the mean square displacement (MSD) of Li⁺ ions was calculated to figure out the diffusion coefficient. Typical linear relationships between MSD and time are plotted in Fig. 4e, which confirms the occurrence of Li⁺ diffusion[52]. The

diffusion coefficients in the bulk crystals were calculated based on Einstein's equation[53], as exhibited in Fig. 4f. The LiF and LiF$_{1-x}$Cl$_x$ crystals possess similar activation energy and Li⁺ diffusion coefficients, which demonstrates the Cl doping has little effect on the Li⁺ transferring through bulk phases, thus confirming that the Li⁺ diffusion along grain boundaries determines the rate of Li⁺ flux across SEI.

## Electrochemical performance of LMA and LMB with dual-halide electrolyte

The cycling behavior of LMA in different electrolytes is presented in Fig. 5. The Li||Cu cells were assembled to measure the CE of LMA in various electrolytes according to Aurbach's method[54,55]. As shown in Fig. 5a, the LMA in BE and LCE suffers from large irreversible capacity, manifested by low CEs of 86.93 and 98.15% due to the ineffective SEI that cannot prevent dead Li formation and solvent decomposition[56,57]. The CE of LMA in HCE reaches ~99.31%, which is benefited from the anion-derived LiF-rich SEI[9,13,28]. With the introduction of DCE in electrolytes, the CE is further improved to 99.54% in 1.3 M LDC electrolyte. This significantly improved CE demonstrates the positive effect of DCE on LMA. 1,1,2,2-tetrachloroethane (TCE) and chlorobenzene (PhCl)

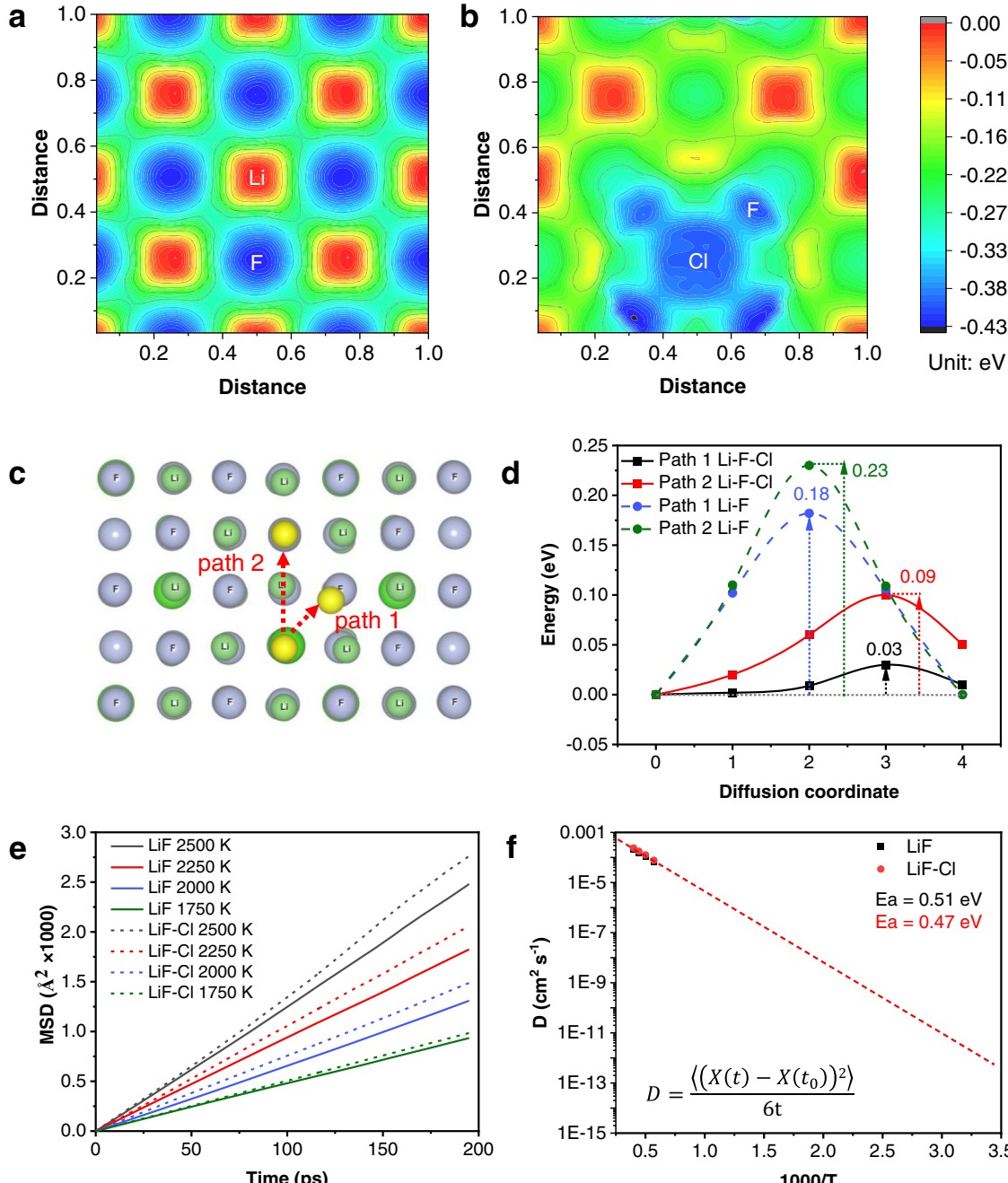

**Fig. 4 | Simulation of Li⁺ transfer in SEI.** Binding energy landscape for Li⁺ diffusion along **a** LiF and **b** LiF$_{1-x}$Cl$_x$ grain boundaries. **c** Schematic illustration of Li⁺ diffusion path. **d** Variation of NEB energies with path 1 and path 2 along LiF or LiF$_{1-x}$Cl$_x$ grain boundaries. **e** Mean square displacement (MSD) of Li⁺ ions in LiF and LiF$_{1-x}$Cl$_x$. **f** Li⁺ diffusion coefficient as a function of temperature. The inserted Einstein's equation expresses the calculation of D from MD simulations, $(X(t) - X(t_0))^2$ is MSD of Li⁺ ions.

were also tested to modulate dual-halide electrolytes (Fig. S18) but display depressed CEs (99.05 and 99.13%), which should be ascribed to the lower LUMO energy that leads to the excessive reduction of PhCl and TCE (Fig. S5). Therefore, appropriate orbital energy is critical for building stable dual-halide SEI. The cycle stability of LMA in 1.3 M LDC was further studied by Li‖Cu cells at 0.5 mA cm⁻² with a fixed capacity of 1 mAh cm⁻², as shown in Fig. 5b. The average CE in 1.3 M LDC displays a fast ramp-up to >99.30% in 50 cycles, signifying the gradual passivation of Cu substrate. With the fully passivated Cu surface, the CE maintains stably at 99.46% over the cycling. Compared with the CE (99.26%) of HCE electrolyte tested by repeated plating/stripping, this is a substantially higher value since the CE is a quantifiable indicator for the lifespan of LMBs[58,59].

The flat Li plating in the dual-halide electrolyte can be reflected by the nucleation overpotential ($\eta_n$)[60,61]. As presented in Fig. 5c, compared to the high $\eta_n$ in BE (184 mV) and HCE (121 mV), the low $\eta_n$ in 1.3 M LDC (88 mV) benefits the emergence of large Li nuclei, which can sustain a flat Li growth and preserve the integrity of native SEI. Furthermore, the plateau overpotential of 1.3 M LDC within the Li growth region is also lower than that in BE and HCE (Fig. 5d) and remains stable with cycling (Fig. S19), suggesting a more favorable Li deposit for the LiF$_{1-x}$Cl$_x$-rich SEI. The morphology of deposited Li metal was confirmed in Fig. 5e, f. Compact Li particles are formed in HCE without obvious Li dendrites. This is attributed to the FSI⁻ derived SEI in which LiF with high surface energy suppresses the growth of dendric Li. However, the rough surface still offers active sites to induce Li dendrites, which is aggravated

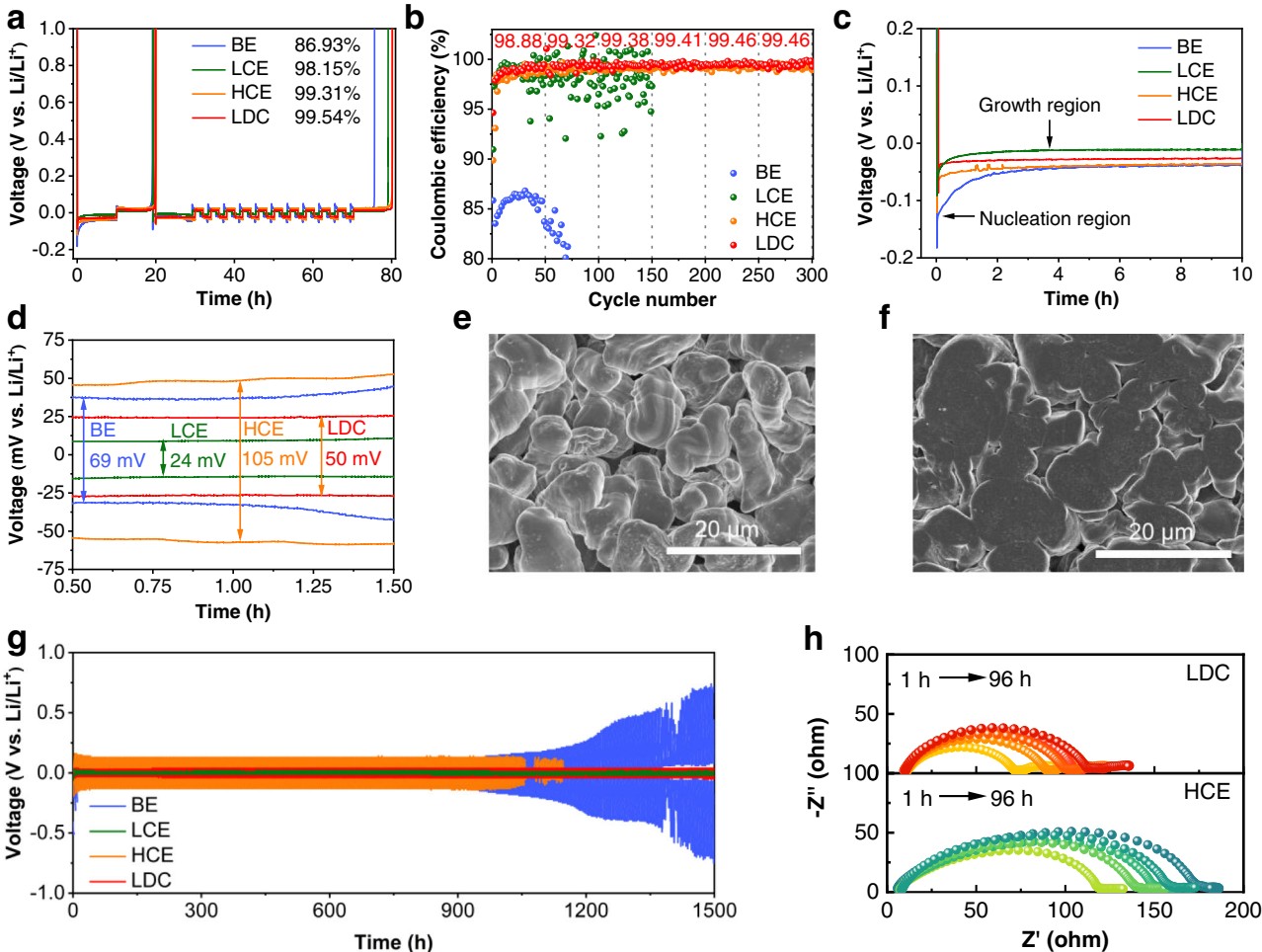

**Fig. 5 | Electrochemical behavior of LMA in different electrolytes.** CE of LMA in Li||Cu cells by **a** using Aurbach's method and **b** repeated plating/stripping (0.5 mA cm⁻²/1 mAh cm⁻²). **c** Voltage profiles of Li plating on Cu at 0.5 mA cm⁻². **d** Overpotentials of Li||Cu cells in the Li growth region. SEM images of deposited Li metal in **e** HCE and **f** LDC electrolytes. **g** Li||Li symmetric cells with different electrolytes were tested at 0.5 mA cm⁻² with a capacity of 1 mAh cm⁻². **h** Evolution of EIS plots in Li||Li cell with LDC and HCE electrolytes.

by the inhomogeneous Li⁺ flux through LiF-rich SEI. For the case of 1.3 M LDC electrolyte, Li is deposited as large nodule-like particles with dense and dendrite-free morphology. Furthermore, the thickness of Li deposited in LDC with a capacity of 4 mAh cm⁻² is 23 μm, which is close to the theoretical value (20 μm) and much thinner than that in HCE (44 μm) (Fig. S20). The flat and dense deposition benefit from the dual-halide SEI that enables spatially homogeneous Li diffusion, thus resulting in smooth Li growth in both horizontal dimension and vertical depth.

Additionally, DCE shows long-term chemical stability to Li foil without distinct bubbling or color change (Fig. S21). Li||Li cells were used to evaluate the long-term cycle stability of LMA (Fig. 5g). The Li||Li cell with the 1.3 M LDC remains stable over 1500 h, while the Li||Li cell with BE and HCE suffers from growing overpotential and short circuit within limited cycles (<1000 h). The high reversibility and stability of LMA in 1.3 M LDC confirm the robustness of LiF$_{1-x}$Cl$_x$-rich SEI. To further evaluate the stability of dual-halide SEI, electrochemical impedance spectroscopy (EIS) of Li||Li cells was conducted as a function of standing time (Fig. 5h). The equivalent circuit fitted by the EIS plots is exhibited in Figure S22a. The semiellipses represent the Li⁺ mobility resistance (R₁) of SEI, which is closely related to electrolyte components. Li||Li cells in 1.3 M LDC electrolyte show a smaller R₁ than that in HCE electrolytes. This is ascribed to the formation of LDC-derived LiF$_{1-x}$Cl$_x$-rich SEI, ensuring faster Li⁺ mobility across SEI. Moreover, the R₁ value in 1.3 M LDC increases slightly, while the R₁

value in HCE displays an obvious increase after 96 h storage (Fig. S22b). This further demonstrates the better stability of the 1.3 M LDC electrolyte to LMA.

As a promising LMB electrolyte, 1.3 M LDC electrolyte was evaluated under harsh conditions using a high-loading NCM811 cathode (≥3.7 mAh cm⁻²) and a thin LMA (20 μm), with an N/P ratio of ~1 (Fig. 6a). The LMBs with BE and LCE electrolytes failed rapidly within 60 cycles due to the low CE of Li metal in BE (Fig. 5b) and the oxidation instability of LCE (Fig. S23a), respectively. Note that the Al corrosion at high voltage is also responsible for the capacity decay in LCE (Fig. S23b–f). Although the HCE prolongs the lifespan to ~130 cycles, the lower Li CE results in the fast depletion of a limited Li source. By comparison, the LMBs using 1.3 M LDC electrolyte realized a long cycle life of over 200 cycles, demonstrating the excellent anodic and cathodic stability of 1.3 M LDC electrolyte. As the cycle proceeds, the voltage profiles of LMBs in 1.3 M LDC become more stable than that in the reference electrolytes (Fig. 6b and S24). Moreover, even using a DMC solvent that is less stable to Li metal anode, the LMBs with an N/P ratio of ~1 still maintained stable for >150 cycles (Fig. S25) in 2.2 M LiFSI/DMC-DCE electrolyte. This is ascribed to the compactness of LiF$_{1-x}$Cl$_x$-rich SEI that prevents the sustainable decomposition of DMC solvents. As for PhCl and TCE-based electrolytes, the assembled Li||NCM811 cells both failed rapidly within 100 cycles (Fig. S26). As a universal high-voltage electrolyte, 1.3 M LDC is also compatible to the high-voltage LCO cathode with an aggressive cutoff-voltage of 4.5 V.

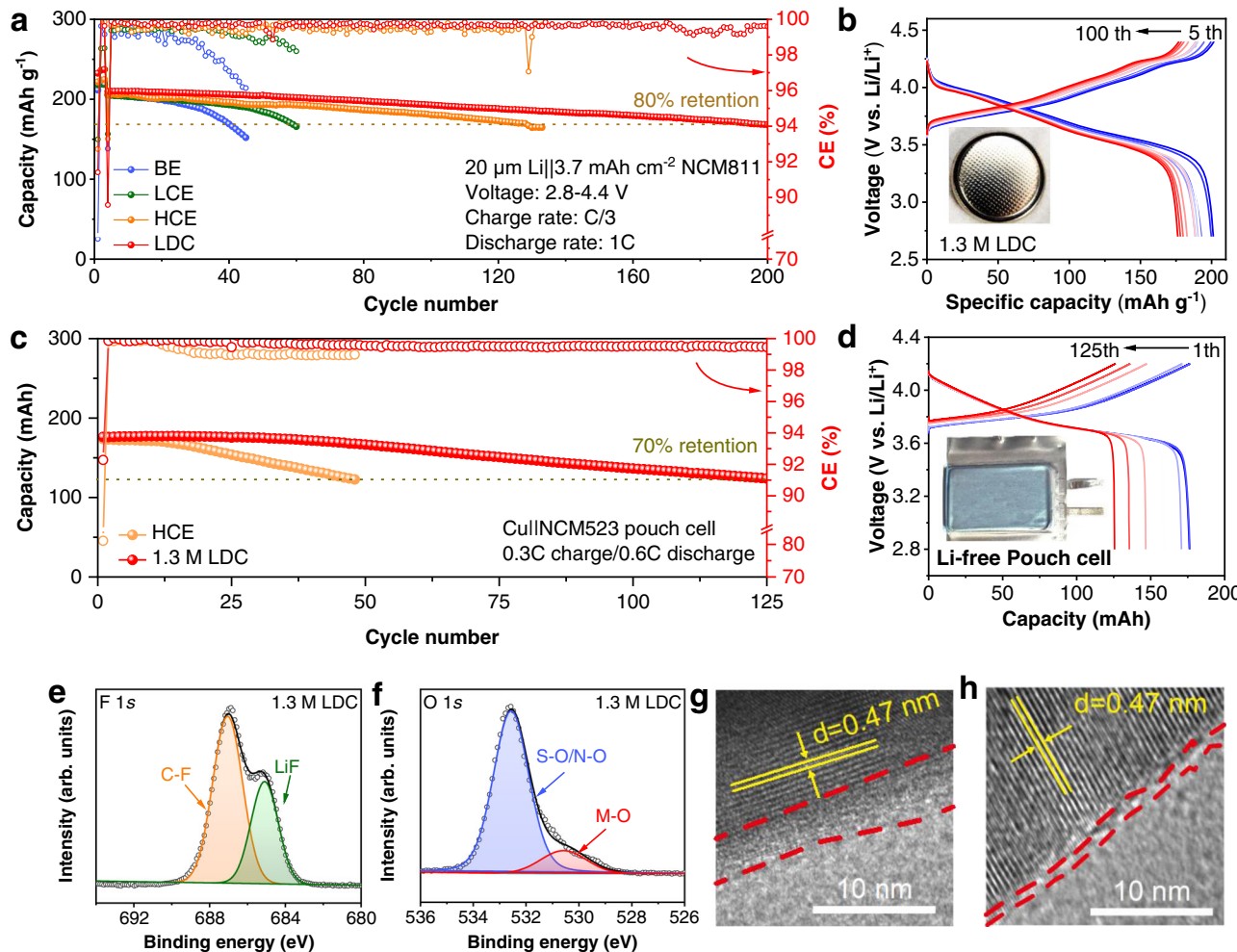

**Fig. 6 | The performance and characterization of LMBs. a** Li||NCM811 cells with different electrolytes (1 C = 200 mA g⁻¹). **b** Voltage profiles of Li||NCM811 with LDC. **c** Cu||NCM523 pouch cells with HCE and LDC (1 C = 180 mA was defined for 180 mAh Cu||NCM523 pouch cells). **d** Voltage profiles of Cu||NCM523 pouch cells with LDC. **e** F 1$s$ and **f** O 1$s$ spectra of NCM811 cycled in 1.3 M LDC electrolytes. HRTEM images of NCM811 cycled in **g** BE and **h** LDC electrolytes.

The Li||LCO cells (20 μm Li, 2 mAh cm⁻² LCO, N/P-2) with 1.3 M LDC yield a high initial capacity of 180 mAh g⁻¹ with high retention of 80% after 240 cycles (Fig. S27). Moreover, the 1.3 M LDC realizes fast Li⁺ diffusion in bulk electrolyte and electrode/electrolyte interfaces, lowering voltage polarization (Fig. S28). As a result, the NCM811 and LCO cathodes both exhibits superior rate capability in 1.3 M LDC electrolytes (Fig. S29).

To better assess the effect of dual-halide electrolytes on full-cell performance, anode-free pouch cells (180 mAh) were cycled at a slow charge (0.3 C) and fast discharge (0.6 C). All the anode-free pouch cells were cycled with 100% depth of discharge. The Cu||NCM523 pouch cells with 1.3 M LDC electrolyte can stably maintain 125 cycles with a retention of 70% (Fig. 6c, d). Furthermore, there are little gassing issues in the Cu||NCM523 pouch cells even without the degassing procedure (Fig. S30). By comparison, Cu||NCM523 pouch cell with HCE failed rapidly due to the low CE and slow Li⁺ diffusion. These results demonstrate the safety and processability of dual-halide electrolyte.

The dual-halide electrolyte also maintains the structural stability of the cathodes (Fig. S31) via the in situ formed robust cathode electrolyte interface (CEI) film. Compared to the surface chemistry in BE (Fig. S32a), a compact inorganic-rich CEI derived from FSI⁻ decomposition covers the NCM811 after cycling in 1.3 M LDC electrolyte (Fig. 6e, f). For NCM811 recovered from BE electrolytes, more organic species were detected (Fig. S32b), such as C-O, C=O, and poly(CO₃²⁻) species. These C-O products are mainly caused by the

dehydrogenation of carbonate solvents[62], which simultaneously generate HF to corrode cathode materials. The structural integrity and the compact CEI were also well supported by high-resolution transmission electron microscopy (HRTEM). As shown in Fig. 6g, h, a thin (~2 nm) and consistent CEI is detected on the NCM811 cycled in 1.3 M LDC electrolyte. However, the BE electrolyte causes a degraded cathode surface due to the continuous reactions between BE electrolyte and NCM811, which impedes Li⁺ transfer and reduces the available capacity of NCM811 as the cycle proceeds, as characterized by the continuously increased impedance in LMBs (Fig. S33). Additionally, the inhomogeneous Li⁺ migration through both cathode and anode interfaces tends to facilitate dendritic Li growth. As a result, the dual-halide electrolyte builds stable interphases at the cathode and anode, which not only maintains the structural stability of high-voltage cathodes but also inhibits Li dendrites by uniform Li⁺ flow.

In summary, through the analysis and identification of Li⁺ transport-driven Li dendrites beneath LiF-rich SEI, we propose a mechanistic protocol for deciphering the correlation between Li⁺ flux and Li growth. An indicator $k$ is defined to reflect the homogeneity of Li⁺ distribution before deposition, which is determined by the local diffusion energy barrier of SEI. '$k \to 0$' is desired to achieve dendrite-free Li deposition under the premise of sufficient ionic conductivity. Additionally, a model dictating the maximum CE that electrolytes can reach is provided to evaluate its compatibility with LMA. These implications guide the design of an effective electrolyte to form the

desired SEI for uniform Li⁺ conduction, as demonstrated by the high CE of 99.54% in Li||Cu cells and flat Li deposition. This strategy sustains the high-voltage Li||NCM811 and Li||LCO full cells over 200 cycles and also enables the anode-free Cu||NCM523 pouch cells with a cycle life of >125 at industrial levels. The successful application of LMBs validates the proposed protocol for exploring and evaluating advanced electrolytes, thus opening up opportunities to enable practical LMBs.

## Methods

### Electrolyte and electrode preparation

All the solvents were purified by molecular sieves prior to use. PhCl, TCE, and DCE solvents were purchased from Sinopharm Chemical Reagent Co., Ltd., J&K Scientific and Aladdin, respectively. The other salts and solvents were purchased from Dadu New Material Co., Ltd. The electrolytes were prepared in an Argon-filled glove box with $O_2$ and $H_2O$ level <0.01 ppm. The solvent ratio of 1 M $LiPF_6$ in ethylene carbonate/dimethyl carbonate (EC/DMC) was set at 1/1 by volume. The other electrolytes (low concentration electrolyte of 1.3 M LiFSI in DME (1.3 M LCE), high-concentration electrolyte of 6 M LiFSI in DME (6 M HCE), 1.3 M LiFSI in DME/DCE (1.3 M LDC), 2.5 M LiFSI in DME/PhCl, 2.4 M LiFSI in DME/TCE, 2.2 M LiFSI in DMC/DCE were prepared according to Table S1 and S2. The thick Li (thickness: 450 μm, area: 1.91 cm⁻²) and thin Li (thickness: 20 μm, area: 1.54 cm⁻²) anodes were obtained from China Energy Lithium Co. Ltd. NCM811 cathodes (area: 1.13 cm⁻²) were prepared by casting the slurry consisting of 96 wt% NCM811, 2 wt% Super P, and 2 wt% polyvinylidene fluoride in N-methyl-2-pyrrolidone onto Al foils, which were calendared after vacuum drying. LCO cathodes (area: 1.13 cm⁻²) were prepared through a similar method to NCM811 cathodes at a weight ratio of 94:2:4 without being calendared. The active material loading of NCM811 and LCO were ~21 and ~13 mg cm⁻², respectively.

### Cell assembly and electrochemical measurements

Polyethylene (PE) was applied as the separator. 2032-type coin cells were assembled for electrochemical tests by using two spacers and adding 100 μL electrolyte. Pouch cells were purchased from LI-FUN Technology Co., Ltd. The electrolyte utilization in pouch cells was 3 g Ah⁻¹. Galvanostatic charge/discharge tests of Li||Cu, Li||Li, Li||NCM811, Li||LCO and anode-free pouch cells were performed on Landt CT 3001 A battery test system. Linear sweep voltammetry (LSV) and Electrochemical Impedance Spectroscopy (EIS) were conducted by an electrochemical station (Ivium). All the cells were kept at 25 °C in a climatic chamber (ShangHai BOLAB Equipment Co., Ltd, BLC-300) for electrochemical tests.

### Material characterizations

Raman spectra were measured by LabRAM HR Evolution with a 532 nm laser. X-ray photoelectron spectroscopy (XPS) spectra were obtained by a Thermo Scientific ESCALAB 250Xi with an Al Ka X-ray source of 1486.6 eV. The microstructure of electrodes was observed by Scanning Electron Microscopy (SEM, Hitachi SU-70) and Transmission electron microscopy (TEM, Tecnai G2 F30).

### Quantum chemistry calculations

The density functional theory (DFT) implanted in Gaussian 09 software was used to perform the quantum chemistry calculations. The equilibrium state structures with geometry optimization were performed by employing the three-parameter empirical formulation B3LYP in conjunction with the basis set of 6−311 + G(d, p). Then the energies of the highest occupied molecular orbital (HOMO) and lowest unoccupied molecular orbital (LUMO) were analyzed.

### Solvation structure simulations

Molecular dynamics (MD) simulations were performed in LAMMPS using the all-atom optimized potentials for liquid simulations (OPLS-AA) force-field with the Li⁺ ions, and FSI⁻ anions description from previous publications[63,64]. The electrolyte systems were set up initially with the salt and solvent molecules distributed in the simulation boxes using Moltemplate (http://www.moltemplate.org/). For each system, an initial energy minimization at 0 K (energy and force tolerances of 10⁻⁵) was performed to obtain the ground-state structure. After this, the system was equilibrated in the constant temperature (298 K) and constant pressure (1 bar) (NpT ensemble) for 5 ns before finally being subjected to 5 ns of constant volume and constant temperature dynamics. Radial distribution functions were obtained using the Visual Molecular Dynamics (VMD) software. Snapshots of the most probable solvation shells were also sampled from the simulation trajectory using VESTA.

### Bulk diffusion for LiF and $LiF_{1-x}Cl_x$

We used the MD method to simulate the lithium diffusion behaviors in the bulk phase of both original and Cl-substituted LiF ($LiF_{1-x}Cl_x$). The structure and crystal lattice parameter for LiF is obtained from the Inorganic Crystal Structure Database (ICSD), while the energy-favorable $LiF_{1-x}Cl_x$ model was filtered from various Cl-doped systems. A $4 \times 4 \times 4$ supercell for LiF and $LiF_{1-x}Cl_x$ was introduced to avoid the imaginary interaction between the unit cells in periodic boundary conditions (PBC). The force-field and corresponding parameters for the Li⁺ and F⁻ ions were obtained from the previous publications[65]. The initial structures were statically relaxed and were set to an initial temperature of 298 K. The structures were then heated to targeted temperatures (1750−2500 K) at a constant rate by velocity scaling over a time period of 1 ps. The NVT ensemble using a Nose-Hoover thermostat was adopted. The total time was set to 500 ps with a time step of 1 fs.

The mean square displacement (MSD) can be used to characterize the diffusion behavior of the system. As in previous studies, the diffusivity $D$ can be calculated based on the following equation:

$$D = \frac{1}{6N\triangle t} \sum_{i=1}^{N} \left\langle |r_i(t + \triangle t) - r_i(t)|^2 \right\rangle_t \quad (5)$$

where $N$ is the total number of diffusion ions, $r_i(t)$ is the position of the $i$-th Li at the time $t$, the diffusion coefficient $D$ can be calculated based on the slope of the MSD curves. The activation energy barrier for Li diffusion can be extracted from the diffusion coefficients at various temperatures according to the Arrhenius equation.

### Surface diffusion for LiF and $LiF_{1-x}Cl_x$

The periodic density functional theory (DFT) calculations were employed to determine dominant diffusion carriers and diffusion pathways, as well as energy barriers of diffusion. Exchange-correlation potentials were parameterized using the generalized gradient approximation (GGA) employing the functional of Perdew−Burke−Ernzerhof (PBE). The projector augmented wave (PAW) approach was used to represent the core electrons and a kinetic energy cutoff of 450 eV was chosen to expand the mono-electronic states in plane waves. The long-range dispersion was accounted for using the DFT-D3 corrections. The self-consistent field (SCF) convergence criterion and the ionic relaxation criterion were set to $1 \times 10^{-6}$ eV and 0.01 eV Å⁻¹, respectively.

$Li_2O$, $Li_2CO_3$, $Li_3N$, and LiF structures were obtained from the Materials Project database. Lattice parameters and atomic positions were then optimized. The $Li_2O$ (111), $Li_2CO_3$ (001), $Li_3N$ (001), and LiF (001) facets were re-optimized. A vacuum of 15 Å was used for each slab to avoid interaction between neighboring slab images. The binding energy landscape of a Li atom on specific surfaces was obtained by scanning the binding energy on various adsorption sites. The climbing image nudged elastic band (CI-NEB) method was employed to study

the diffusion of lithium on slabs, aiming to locate the transition states and verify the minimum energy path.

## COMSOL simulation

It is assumed that the SEI film uniformly and stably covers the anode and cathode. The thickness and ion conductivity of SEI are invariable during the deposition process. Two different electrolytes, i.e., HCE and LDC in this work, are adopted for comparison. Due to the considerable magnitude difference between SEI thickness (~10 nm) and electrolyte thickness (~25 μm), the modeling for electrodeposition of Li-ion is realized by simulating mass transfer in liquid electrolyte and SEI successively. The necessary parameters for modeling are listed in Table S3.

The mass flux of Li ions in electrolyte and SEI is given by Nernst–Planck equation

$$\begin{cases} \overrightarrow{N}_E = -D_E \nabla c_E - nFt_E c_E \nabla \varphi_E \\ \overrightarrow{N}_S = -D_S \nabla c_S - nFt_S c_S \nabla \varphi_S \end{cases} \quad (6)$$

The mass transfer equation

$$\frac{\partial c}{\partial t} + \nabla \cdot \overrightarrow{N} = 0 \quad (7)$$

The local current density on anode or cathode as a function of potential $\varphi$, Li-ion concentration $c$ can be expressed by

$$i = F(k_a)^{\alpha_c} (k_c)^{\alpha_a} \left( \frac{c}{c_{ref}} \right)^{\alpha_a} \left[ \exp\left( \frac{\alpha_a F \eta}{RT} \right) - \exp\left( \frac{-\alpha_c F \eta}{RT} \right) \right] \quad (8)$$

Where $\alpha_c$ and $\alpha_a$ are the cathodic and anodic transfer coefficients, respectively, and $\alpha_c$ and $\alpha_a$ for a single-electron reaction. $\eta$ is the overpotential that can be expressed as $\eta = \varphi_S - \Delta\varphi_{film} - \varphi_l - E_{eq}$, where $\varphi_S$ is the exerted potential on the Li electrode, $\varphi_l$ is the local potential, $\Delta\varphi_{film}$ is the film electronic resistance and $E_e$ is the equilibrium potential of reaction. It is noting that $\Delta\varphi_{film} = \frac{\delta}{\sigma_{SEI}}$ is only utilized in the electrolyte region to consider the SEI effect.

COMSOL Multiphysics 5.4 platform is used to establish the above model and to numerically solve it. The sizes of the simulation area are 120 μm × 25 μm for the electrolyte region, 80 nm × 10 nm for SEI of HCE electrolyte, and 80 nm × 8.5 nm for SEI of LDC electrolyte. A fine mesh is adopted with the maximum grid size of 1 μm and 1 nm for simulating the electrolyte and SEI region, respectively.

## Data availability

The source data that support the findings of this study are available from the corresponding author upon reasonable request.

## Code availability

The codes that support the findings of this study are available from the corresponding author upon reasonable request.

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

## Acknowledgements

This work is supported by the National Natural Science Foundation of China (22072134, 22161142017, and U21A2081), Natural Science Foundation of Zhejiang Province (LZ21B030002), the Fundamental Research Funds for the Zhejiang Provincial Universities (2021XZZX010), the Fundamental Research Funds for the Central Universities (2021FZZX001-09), and "Hundred Talents Program" of Zhejiang University.

## Author contributions

S.Z., R.L., and N.H. contributed equally to this work, they analyzed the results and wrote the manuscript. S.Z., L.F., and X.F. conceived the idea. S.Z. performed the material characterization and electrochemical measurements. R.L. designed the crystal configurations and performed the theoretical calculations. N.H. established the mechanistic model and

conducted the COMSOL simulations. S.W. and X.W helped measure the TEM images of cathodes. T.D., Z.W., D.L., H.Z., J.Z., L.C., L.F., and X.F. participated in the discussions. X.F. supervised all the studies.

## Competing interests

The authors declare no competing interests.
