## [Peer Review File · Nature Communications]

Reviewer comments , first round review -

Reviewer #1 (Remarks to the Author):

Using 1,2-dichloroethane as a co-solvent, these authors propose and experimentally validate a dual-halide (LiF_{1-x}Cl_x) SEI, which benefits to flat Li plating and stably cycling of Li anode. The work is of high interest, suitable for publication after clarifying the questions and concerns raised below:

1. What is the long-term chemical stability between DCE and Li, and the effect of FSI⁻ and Cl⁻ anions on the corrosion of cathode current collector (Al).
2. L273, "The LiF and LiF_{1-x}Cl_x crystals possess similar activation energy and Li⁺ diffusion coefficients", are there any theoretical supports that LiF and LiCl form a LiF_{1-x}Cl_x crystal (i.e., Cl⁻ replaces F⁻ in the LiF cell lattice), other than a mixture or composite, (1-x)LiF-xLiCl.
3. "mobility" would be a better terminology than "permeability".
4. "HCE" in Figure 3a seems to be a typo (LiFSI?). If yes, "the free FSI⁻ anions at 717 cm⁻¹ blueshifts to 752 cm⁻¹" in L207 should be corrected as "redshifts from 752 cm⁻¹ to 717 cm⁻¹".
5. Mention and explain why the LDC cell has a lower bulk resistance compared with the HCE cell in Figure 5h.
6. What is the ratio (vol. or wt.) of DCE in LDC and how does DCE affect LiFSI solubility and the ionic conductivity of electrolyte?

Reviewer #2 (Remarks to the Author):

The manuscript presents a design principle for achieving Li plating and minimization of dendrite growth in Li-metal batteries. The manuscript presents comprehensive modeling and extensive characterization analysis to demonstrate the proposed mechanistic concept. Authors recognize that one of the key reasons for Li-dendrite growth is heterogeneity in SEI structure and non-uniform Li⁺ permeability through SEI. Therefore, they propose a new electrolyte that allows formation of dual-halide (F and Cl) SEI to improve homogeneity in Li⁺ transport through SEI. As a result they demonstrate that cells developed following this principle show great cycling ability. The manuscript is clearly written, well organized. More importantly, compared to many battery topic papers, this manuscript has a well-formulated fundamental principle/mechanism, extensive theoretical support illustrating the mechanism, and proof-of-the-concept validation through experiments. As such, I believe this manuscript stands out from many other manuscripts in this area and can be recommended for publication.

Couple minor comments:

- 1) Authors need to be careful when relating solvation structure of Li⁺ in bulk electrolyte to possible SEI composition or redox pathway. The composition of ions' solvating shell near charged electrode surface can significantly differ from its bulk composition. The electric double layer composition can strongly depend on electrode voltage and hence influence the composition of SEI (see e.g. papers by Borodin et al.).
- 2) Not sure what is the purpose/benefit of showing number of Li around FSI in Fig.3e. Most of the discussion and Fig. 3f are focused on Li solvation. I think it would be more informative to summarize composition of Li coordination shell. Ion clustering can be still illustrated with Li-Li pdf and coordination.

Point-by-point Response to the Reviewers' Comments

Nature Communications manuscript NCOMMS-22-24251A

Title: Tackling realistic Li⁺ flux for high-energy Li metal batteries

Reviewer #1

Using 1,2-dichloroethane as a co-solvent, these authors propose and experimentally validate a dual-halide (LiF_{1-x}Cl_x) SEI, which benefits to flat Li plating and stably cycling of Li anode. The work is of high interest, suitable for publication after clarifying the questions and concerns raised below.

Reply: Thanks for the reviewer's positive comment on our work.

1. What is the long-term chemical stability between DCE and Li, and the effect of FSI⁻ and Cl⁻ anions on the corrosion of cathode current collector (Al).

Reply: Thanks for the reviewer's great comment.

We have evaluated the long-term chemical stability between DCE and Li by soaking a Li foil in DCE solvent. After storage at 25 °C for one week, no obvious side reactions were observed.

In the revised manuscript, we added:

Additionally, DCE shows long-term chemical stability to Li foil without distinct bubbling or color change (Figure S21).

In the revised supporting information, we added:

Figure S21. Comparison of optical images of a Li foil soaking in DCE for 1 week

To evaluate the effect of FSI⁻ and Cl⁻ anions on the corrosion of Al, Li||Al cells with different electrolyte were held at 4.5 V for 12 h. The SEM images of Al foils are shown in the Figure S23.

In the revised manuscript, we added:

Note that the Al corrosion at high voltage is also responsible for the capacity decay in LCE (Figure S23b-23f).

In the revised supporting information, we added:

Figure S23. (a) LSV curves of different electrolytes tested in Li||Al cells. SEM images of (b) original Al foils and Al foils held at 4.5 V for 12 h in (c) BE, (d) 1.3 M LCE, (e) HCE and (f) 1.3 M LDC electrolytes, respectively.

Li||cells were held at 4.5 V for 12 h to measure the corrosivity of different electrolytes to Al foils. Compared to the original Al foil (Figure S23b), BE electrolyte shows no corrosivity to the Al foil (Figure S23c), while the 1.3 M LCE obviously corrodes the Al foil due to the free FSI⁻ anions (Figure S23d). However, the Al corrosion is significantly inhibited in HCE (Figure S23e) and 1.3 M LDC electrolyte (Figure S23f).

2. L273, “The LiF and LiF_{1-x}Cl_x crystals possess similar activation energy and Li⁺ diffusion coefficients”, are there any theoretical supports that LiF and LiCl form a LiF_{1-x}Cl_x crystal (i.e., Cl⁻ replaces F⁻ in the LiF cell lattice), other than a mixture or composite, (1-x)LiF-xLiCl.

Reply: Thanks for the reviewer's constructive comment.

To offer the theoretical supports that LiF and LiCl form a $\text{LiF}_{1-x}\text{Cl}_x$ crystal (i.e., Cl⁻ replaces F⁻ in the LiF cell lattice), other than a mixture or composite, $(1-x)\text{LiF}-x\text{LiCl}$, we have added "energy above hull" into the Figure S17u, which also confirms the thermodynamic feasibility of forming $\text{LiF}_{1-x}\text{Cl}_x$ crystal.

In the revised manuscript, we added:

According to the formation energy and energy above hull summarized in Figure S17u, the $\text{LiF}_{1-x}\text{Cl}_x$ configuration in Figure S17t is the most stable state, in which some of F atoms are randomly replaced by Cl atoms.

In the revised supporting information, we added:

Figure S17. (a-t) Possible configurations of $\text{LiF}_{1-x}\text{Cl}_x$ (Li: green balls, F: gray balls, Cl: red balls).

(u) Formation energy and energy above hull of each $\text{LiF}_{1-x}\text{Cl}_x$ configuration.

The thermodynamic feasibility of each $\text{LiF}_{1-x}\text{Cl}_x$ configuration was estimated by the formation energy and energy above hull (E_{hull}) in Figure S17u. E_{hull} of $\text{LiF}_{1-x}\text{Cl}_x$ is the relative formation energy (E_f) against segregation into thermodynamically stable phases

of LiF and LiCl: $E_{\text{hull}} = E(\text{LiF}_{1-x}\text{Cl}_x) - (1-x)E(\text{LiF}) - xE(\text{LiCl})$. The lowest E_f and E_{hull} both prove that the configuration in Figure S17t is the most stable state. This also demonstrates LiF and LiCl form a $\text{LiF}_{1-x}\text{Cl}_x$ crystal, other than a mixture or composite of $(1-x)\text{LiF} - x\text{LiCl}$.

3. “mobility” would be a better terminology than “permeability”.

Reply: Thanks for the reviewer’s great suggestion, which improved the quality of our work.

We have replaced “permeability” by “mobility” in the main text and Figures of the revised manuscript and revised supporting information.

4. “HCE” in Figure 3a seems to be a typo (LiFSI?). If yes, “the free FSI⁻ anions at 717 cm^{-1} blueshifts to 752 cm^{-1} ” in L207 should be corrected as “redshifts from 752 cm^{-1} to 717 cm^{-1} ”.

Reply: We really appreciate the reviewer’s comment.

“HCE” in Figure 3a is not a typo. The “HCE” in Figure 3a is the electrolyte of 6 M LiFSI-DME. The high ratio of LiFSI salt to DME solvent in HCE results in the similar Raman spectra to that of LiFSI.

5. Mention and explain why the LDC cell has a lower bulk resistance compared with the HCE cell in Figure 5h.

Reply: Thanks for the reviewer’s great suggestion.

We have supplemented and explained why the LDC cell has a lower bulk resistance compared with the HCE cell in the revised manuscript (Page 19):

To further evaluate the stability of dual-halide SEI, electrochemical impedance spectroscopy (EIS) of Li||Li cells was conducted as a function of standing time (Figure 5h). The equivalent circuit fitted by the EIS plots is exhibited in Figure S22a. The semiellipses represent the Li⁺ mobility resistance (R_1) of SEI, which is closely related to electrolyte components. Li||Li cells in 1.3 M LDC electrolyte show a smaller R_1 than

that in HCE electrolyte. This is ascribed to the formation of LDC-derived $\text{LiF}_{1-x}\text{Cl}_x$ -rich SEI, ensuring the faster Li^+ mobility across SEI.

6. What is the ratio (vol. or wt.) of DCE in LDC and how does DCE affect LiFSI solubility and the ionic conductivity of electrolyte?

Reply: We really appreciate the reviewer's constructive comment.

We have added the volume ratio in Table S2 in the revised supporting information.

Table S2 Detailed information of the dual-halide electrolytes

Electrolytes	Molar ratio	Volume ratio	Density (g cm^{-3})	Molarity (mol L^{-1})
LiFSI/DME-DCE	0.949/1/6	1/4.531	1.193	1.3
LiFSI/DME-PhCl	0.806/1/2	1/1.951	1.270	2.5
LiFSI/DME-TCE	0.811/1/2	1/2.026	1.495	2.4
LiFSI/DMC-DCE	0.696/1/2	1/1.904	1.312	2.2

The effects of DCE on LiFSI solubility and the ionic conductivity of electrolyte have been shown in Figure S4 in the revised supporting information.

Figure S4. (a) Li^+ conductivity and (b) viscosity of 1 M $\text{LiPF}_6/\text{EC-DMC}$ (BE), 1.3 M LiFSI/DME-DCE (1.3 M LDC), 6 M LiFSI/DME (HCE) electrolytes, LiFSI/DME-TTE^2 , LiFSI/DMC-TTE^3 and LiFSI/DME-FB^4 . (c) Variation of Li^+ concentration and conductivity with the DME/DCE molar ratio.

Figure S4c displays the variation of Li^+ concentration and conductivity with the DME/DCE molar ratio. 1.3 M LDC electrolyte was selected for further investigation due to its high conductivity and comparable concentration to commercial electrolyte.

Reviewer #2:

The manuscript presents a design principle for achieving Li plating and minimization of dendrite growth in Li-metal batteries. The manuscript presents comprehensive modeling and extensive characterization analysis to demonstrate the proposed mechanistic concept. Authors recognize that one of the key reasons for Li-dendrite growth is heterogeneity in SEI structure and non-uniform Li^+ permeability through SEI. Therefore, they propose a new electrolyte that allows formation of dual-halide (F and Cl) SEI to improve homogeneity in Li^+ transport through SEI. As a result, they demonstrate that cells developed following this principle show great cycling ability. The manuscript is clearly written, well organized. More importantly, compared to many battery topic papers, this manuscript has a well-formulated fundamental principle/mechanism, extensive theoretical support illustrating the mechanism, and proof-of-the-concept validation through experiments. As such, I believe this manuscript stands out from many other manuscripts in this area and can be recommended for publication.

Reply: Thanks for the reviewer's positive comment on our manuscript.

1) Authors need to be careful when relating solvation structure of Li^+ in bulk electrolyte to possible SEI composition or redox pathway. The composition of ions' solvating shell near charged electrode surface can significantly differ from its bulk composition. The electric double layer composition can strongly depend on electrode voltage and hence

influence the composition of SEI (see *e.g.* papers by Borodin et al.).

Reply: Thanks very much for your great suggestions.

We fully agree with the above reviewer's opinion that electrode voltage can affect the electric double layer composition and further influence the SEI composition. To supplement the discussion about ions' solvating shell and SEI formation, we carried out AIMD simulation of interphase evolution between Li anode and electrolyte in Figure S10, which reveals the formation of $\text{LiF}_{1-x}\text{Cl}_x$ specie in SEI.

In the revised manuscript, we added:

These phenomena indicate the preferential decomposition of FSI⁻ anions in 1.3 M LDC electrolyte, accompanying by the DCE decomposition (Figure S5) to produce $\text{LiF}_{1-x}\text{Cl}_x$ species, which is demonstrated by the *ab initio* MD in Figure S10.

In the revised supporting information, we added:

Figure S10. Evolution of interphase between LMA and 1.3 M LDC electrolyte: (a) 0 fs, (b) 200 fs and (c) 1000 fs.

Figure S10 illustrates the *ab initio* MD results of interphase between LMA and 1.3 M LDC electrolyte, where line model represents the unreacted components and ball-stick model represents the reduction products. After 1000 fs simulation, the $\text{LiF}_{1-x}\text{Cl}_x$ cluster can be observed at the LMA surface.

2) Not sure what is the purpose/benefit of showing number of Li around FSI in Fig.3e. Most of the discussion and Fig. 3f are focused on Li salvation. I think it would be more informative to summarize composition of Li coordination shell. Ion clustering can be still illustrated with Li-Li pdf and coordination.

Reply: We really appreciate the reviewer's constructive comment.

The purpose/benefit of showing number of Li around FSI in Fig.3e is to conveniently display the proportion of CIP (anion coordinating to a single Li^+ cation) and AGG (anion coordinating to two or more Li^+ cations) clusters in the electrolyte, which generate anion-derived SEI.

Moreover, we heartily agree with the review's view that summarizing composition of Li coordination shell is more informative. Accordingly, we have added the Li-Li pdf and illustrated the representative ion clusters in the Figure S9.

In the revised manuscript, we added:

Furthermore, radial Li-Li pair distribution function was calculated and analyzed in Figure S9, where ion clusters with size of 6 Å account for the largest proportion.

In the revised supporting information, we added:

Figure S9. (a) Radial Li-Li pair distribution function, (b) statistical results of the ion clusters of 1.3 M LDC electrolyte. (c) Representative ion clusters consisting of 6 Li^+ ions and 6 FSI^- anions.

Figure S9a displays the radial Li-Li pair distribution function. The sharp peak at 6 Å indicates the large ion clusters of AGGs. Figure S9b exhibits the statistical results of ion clusters. The ion clusters consisting of 6 Li^+ /6 FSI^- and 10 Li^+ /10 FSI^- account for the largest proportion, confirming the domination of AGGs in 1.3 M LDC electrolyte. Figure S9c illustrates a representative ion cluster including 6 Li^+ ions and 6 FSI^- anions.

Reviewer comments , second round review -

Reviewer #1 (Remarks to the Author):

My concerns have been well addressed, no further revision is needed.